# Bacteriophage Therapy to Reduce Colonization of *Campylobacter* *jejuni* in Broiler Chickens before Slaughter

**DOI:** 10.3390/v13081428

**Published:** 2021-07-22

**Authors:** Daniela D’Angelantonio, Silvia Scattolini, Arianna Boni, Diana Neri, Gabriella Di Serafino, Philippa Connerton, Ian Connerton, Francesco Pomilio, Elisabetta Di Giannatale, Giacomo Migliorati, Giuseppe Aprea

**Affiliations:** 1Istituto Zooprofilattico Sperimentale dell’Abruzzo e del Molise “G. Caporale”, 64100 Teramo, Italy; d.dangelantonio@izs.it (D.D.); s.scattolini@izs.it (S.S.); f.pomilio@izs.it (F.P.); e.digiannatale@izs.it (E.D.G.); g.migliorati@izs.it (G.M.); 2Istituto Superiore di Sanità, 00161 Rome, Italy; arianna.boni@iss.it; 3Local Health Unit of Ferrara (USL Ferrara), 44121 Ferrara, Italy; an.aid@live.it; 4Local Health Unit of Lanciano-vasto-Chieti (ASL Chieti), 66100 Chieti, Italy; g.diserafino@aslchieti.it; 5Division of Food Science, School of Bioscience, The University of Nottingham, Nottingham LE12 5RD, UK; Pippa.Connerton@nottingham.ac.uk (P.C.); ian.connerton@nottingham.ac.uk (I.C.)

**Keywords:** anti-microbial resistance (AMR), broiler chickens, *Campylobacter jejuni*, phage therapy

## Abstract

Campylobacteriosis is the most commonly reported gastrointestinal disease in humans. *Campybacter jejuni* is the main cause of the infection, and bacterial colonization in broiler chickens is widespread and difficult to prevent, leading to high risk of occurrence in broiler meat. Phage therapy represents an alternative strategy to control *Campylobacter* in poultry. The aim of this work was to assess the efficacy of two field-isolated bacteriophages against experimental infections with an anti-microbial resistant (AMR) *Campylobacter* *jejuni* strain. A two-step phage application was tested according to a specific combination between chickens’ rearing time and specific multiplicities of infections (MOIs), in order to reduce the *Campylobacter* load in the animals at slaughtering and to limit the development of phage-resistant mutants. In particular, 75 broilers were divided into three groups (A, B and C), and phages were administered to animals of groups B and C at day 38 (Φ 16-izsam) and 39 (Φ 7-izsam) at MOI 0.1 (group B) and 1 (group C). All broilers were euthanized at day 40, and *Campylobacter* *jejuni* was enumerated in cecal contents. Reductions in *Campylobacter* counts were statistically significant in both group B (1 log_10_ colony forming units (cfu)/gram (gr)) and group C (2 log_10_ cfu/gr), compared to the control group. Our findings provide evidence about the ability of phage therapy to reduce the *Campylobacter* load in poultry before slaughtering, also associated with anti-microbial resistance pattern.

## 1. Introduction

*Campylobacter* is the most commonly reported gastrointestinal bacterial pathogen in humans in the European Union (EU) since 2005, with 220,682 confirmed cases of Campylobacteriosis in 2019 [1]. *Campylobacter* spp., in particular *Campylobacter jejuni* (*C. jejuni*) and *Campylobacter coli* (*C. coli*), are ubiquitous in nature, and their niche seems to be the intestinal mucosa of warm-blooded hosts [2,3], especially those of avian species. In poultry, the natural bacterial colonization is mainly associated with horizontal transmission at the farm level, and to a lesser extent to vertical transmission [4], with a mean prevalence in the EU of 71.2% of broiler batches [5]. While *Campylobacter* in chickens does not cause any visible symptoms, in humans some strains can lead to severe disease, with clinical signs like fever, headache, abdominal pain and diarrhea, nausea, and vomiting. Most cases of Campylobacteriosis are self-limiting, but there is the potential of post-infection complications such as Guillain-Barre syndrome, irritable bowel syndrome, and septicemia [6]. Animal-to-human transmission is mostly related to the food route: the consumption of undercooked poultry meat and the cross-contamination from other foods during the preparation of poultry meat are significant risk factors [7]. However, human infections are also caused by *Campylobacter* isolates originating from cattle or other animals [8,9,10], while person-to-person transmission is considered relatively uncommon [11].

Nowadays, the increasing problem of multi-drug resistance to antibiotics largely used in animals and humans, and in particular to treat *C. jejuni* infections, is considered a serious threat to public health [12]. Both the global burden related to anti-microbial resistance (AMR) [13] and the increasing trends of hospitalizations for many zoonotic bacteria such as *Campylobacter* spp. [14] have clear public health and economic implications. In this scenario, it is well known that the farm level plays a key role regarding the entrance of *Campylobacter* into the food chain, thus leading to the risk for consumers to be infected, especially via poultry meat consumption [15]. The EU summary report on AMR in zoonotic and indicator bacteria from humans, animals, and foods underlines the worrying levels of resistance to ciprofloxacin, nalidixic acid, and tetracycline [13]. According to an opinion from the European Food Safety Authority (EFSA) [16], the reduction of *Campylobacter* counts by more than 2 log_10_ units in meat could reduce the public health risk for human consumers to contract Campylobacteriosis after the consumption of chicken meals by more than 90%.

For these reasons, studies are necessary to find more alternatives and safe measures to reduce *Campylobacter* spp. loads in chickens. Intervention strategies such as feed and water additives, vaccines, and biosecurity measures have been deeply investigated [17]. Recently, researchers have returned to bacteriophages as promising and welfare-friendly tools for the biological control of *Campylobacter* spp. on farms [16]. Among the innovative strategies also suggested by EFSA experts, bacteriophage-based applications in chickens could play an important role in reducing the load of *Campylobacter* spp. in the birds [16]. Bacteriophages (or phages) are the most abundant biological entities on earth [18,19], and their efficacy to reduce *Campylobacter* spp. in broiler chicken ceca [20] and on the surface of chicken skin [21] has already been demonstrated. The data of a recent study showed that exposure of the rat gut microbiome to a cocktail of commercial phage preparations active against *Salmonella enterica*, *Staphylococcus aureus*, *Streptococcus pyogenes*, *Proteus* (*P.*) *mirabilis*, *P. vulgaris*, *Pseudomonas aeruginosa*, *Klebsiella pneumoniae*, and *Escherichia coli* resulted in dysbiosis with increased inflammation and gut permeability [22]. However, *Campylobacter* bacteriophage applications in poultry demonstrated the ability to reduce *C. jejuni* counts without affecting the microbiota structure of the animals [23]. The key to a meaningful intervention within flocks is based on a mix of factors that range from the selection of efficient phages and the use of specific multiplicities of infection (MOIs) to methods and timing of phage-based applications in poultry [24].

Moreover, another problem encountered in phage therapy is related to the development of phage-resistant bacterial mutants [25] that could render any phage-based applications unsuccessful. In order to avoid this problem, many authors, instead of using only one phage in their in vivo trials [20,26], suggested the use of phage cocktails [23,27] instead.

The aim of this research was to assess the in vivo efficacy of two field bacteriophages to reduce the load of an experimental AMR *Campylobacter jejuni* strain in broilers before slaughtering. To achieve meaningful results in terms of *Campylobacter* spp. reduction, the authors worked especially on two specific aspects: the timing of phage administration in chickens and the MOIs to apply. In particular, in our study a different and innovative approach not used before, based on a sequential application of two different phages (specifically isolated from poultry farms) administrated individually after a 24-h interval, one day before slaughtering, was adopted in order to limit any potential occurrence of phage-resistant *Campylobacter* strains [25]. Moreover, because of the worldwide growing concern about AMR, a specific *Campylobacter* strain (252 gM/12A) was used in this study, chosen on the basis of its anti-microbial resistance pattern against some antibiotics frequently used in veterinary and human medicine (ciprofloxacin and nalidixic acid) [28]. The preliminary results reported in this paper are extremely encouraging; in fact, the reduction of *Campylobacter* loads in broilers of 2 log_10_ units would reduce the public health risk for the human consumers to get Campylobacteriosis by eating contaminated chicken meals by up to 90% [16]. Moreover, the two field bacteriophages administered to poultry before slaughtering showed activity against a specific AMR *Campylobacter* strain, and this was another important finding that deserves more in-depth investigations, also with the support of specific field studies to be carried out in the near future.

## 2. Materials and Methods

### 2.1. Campylobacter Strains and Bacteriophages

The bacterial strain used for chicken infection in the trial (*C. jejuni* 252 gM/12A) was isolated from poultry farms and characterized within the Italian Reference Laboratory for *Campylobacter*. This specific strain was chosen among the others because of its particular sensitivity to bacteriophages and for its AMR pattern against ciprofloxacin and nalidixic acid, antibiotics frequently used in veterinary and human medicine [28]. The strain was sub-cultured from the original frozen cryogenic bead stocks and incubated under micro-aerobic conditions (85% nitrogen, 5% oxygen, and 10% carbon dioxide) at 42 °C for 48 h (h), according to the International Standard Organization (ISO) method 10272–1 (2006). To achieve the concentrations suitable for the trial (10^5^ colony-forming unit (cfu)/milliliter (mL)), bacterial cell suspensions were diluted in 10 mL of phosphate-buffered saline (PBS; Dulbecco’s Formula Modified, ICN Biochemicals, Thame, Oxfordshire, UK).

The bacteriophages used in this study, Φ7-izsam and Φ16-izsam, were previously isolated and morphologically characterized, as reported by Aprea et al. (2018) [29]. In particular, these phages were ascribed to the *Caudovirales* order, *Myoviridae* family, Group A, thus belonging to double-stranded DNA viruses [29], and were chosen for their wide host-range spectrum and their particular activity against AMR *C. jejuni* strains [28]. Before the experiment, the two bacteriophages were subjected to a three-step plaque purification assay, in order to assure their purity. Briefly, for each purification step, the plaques were individually picked up from the agar and eluted into 2 mL of SM buffer (0.05 M TRIS, 0.1 M NaCl, 0.008 M MgSO_4_, 0.01% (weight in volume) gelatin, pH 7.5) for 8–9 h at room temperature; then the supernatant was filtered with 0.45 micrometer (μm) filters. Subsequently, 1 mL of *Campylobacter* broth culture (252 gM/12A field strain) was added to 1 mL of the supernatant and left for 15 min at 37 ± 1 °C in aerophilic conditions, to allow phages to adsorb on the bacterial cells. At the end of the incubation period, the culture was added to 4 mL of New Zealand Casamino Yeast Medium (NZCYM) soft agar (NZCYM broth + 7 grams (g)/liter (L) of agar - Life Technologies, Milan, Italy), poured on a plate of NZCYM agar (NZCYM broth + 15 g/L agar - Life Technologies, Milan, Italy), and incubated at 37 ± 1 °C in micro-aerophilic conditions for 48 h. After the incubation, the plate was coated with 5 mL of SM buffer. The soft agar was gently fragmented with a sterile loop and left to elute for 6 h at room temperature under gentle agitation (60 run per minute (rpm)). The supernatant was then filtered with 0.45 μm filters and subjected to titration. This procedure was repeated three times. The two bacteriophages were finally subjected to further replication steps for increasing their titers, individually [17]. To reach the final concentration loads (10^7^ plaque-forming unit (pfu)/mL for MOI 0.1 and 10^8^ pfu/mL for MOI 1) for the in vivo trial, phage suspensions were prepared via dilutions in SM broth, with the addition of 30% *w/v* of calcium carbonate (CaCO_3_), to prevent phage inactivation by the acidity of the chickens’ stomachs.

### 2.2. Broiler Rearing and In Vivo Trial

For the in vivo trial, the Ross 308 broiler chickens were chosen. The number of animals was reduced to the minimum required to ensure statistically positive results: in total, 75 chickens were enrolled in the assay. The animals were housed in an environment with enrichments in order to stimulate motor activity for their welfare. In the shelter room, infrared lamps were used to provide the optimal temperature for the animals (Table 1).

The animals were placed on a soft surface to ensure the manifestation of strictly specific behaviors such as fluttering wings and distortion of the paws. The relative humidity (RU) was adjusted according to the age of the animals. In particular, from day 1 to day 4 of life, RU was maintained at 65/70%, and from day 5 to day 37 of life, at 60%. The animals were exposed to 24 h of light for the 1st week of life, following a natural day/night cycle from the 2nd week of life to the end of the experiment. The animals were reared in a single group from day 1 to day 36, at the feeding conditions described in Table 2. Both environmental temperatures (Table 1) and feed formulations (Table 2) were adapted from the standard rearing conditions.

The experimental phases of the in vivo trial are summarized in Table 3.

In particular, at day 0, soon before the experimental infection, cloacal swabs from birds were assessed in order to check the birds for the absence of natural *Campylobacter* spp. infections (ISO 10272: 2006 Part 1). For 1-day chicks, *C. jejuni* 252 gM/12A (0.1 mL of the bacterial suspension 10^5^ cfu/mL) was administered via oral gavage using a sterile 5 mL volume syringe and a detachable 6 cm-long cannula. At day 10, cloacal swabs were taken using swabs (Medical Wire & Equipment: MW171) and the *Campylobacter* colonization was verified, using standard accredited methods (ISO 10272: 2006 Part 1 and Part 2). At day 30, a statistically significant number of birds (n = 6) were sacrificed, and the *Campylobacter* count in their gut content was assessed and recorded in order to set up the phage suspensions at MOI 0.1 and 1. At day 37, the animals that survived were randomly subdivided into three experimental groups (A, B, and C) of 23 animals each. The day after (day 38), birds of group A, representing the negative control, were treated with SM broth with the addition of 30% (*w/v*) CaCO_3_ (SM-CaCO_3_). Bacteriophages were administered to chickens in groups B and C via oral gavage in the following conditions: experimental group B chicks were orally treated with ϕ16-izsam at MOI 0.1 (phage suspension of 10^7^ pfu/mL); and experimental group C chicks were treated with Φ16-izsam at MOI 1 (phage suspension of 10^8^ pfu/mL). The next day (day 39), the other phage, Φ7-izsam, was administered at MOI 0.1 to animals of group B and at MOI 1 to animals of group C. The animals of group A received SM-CaCO_3_ only. At day 40, all the birds from the three groups were sacrificed and *C. jejuni* loads were assessed from cecal contents (ISO 10272: 2006 Part 1 and Part 2). In particular, 40 days of life is the age at which animals are industrially slaughtered to reach a commercial weight of about 2.8 kg. The results among the experimental groups were statistically analyzed to verify significant differences within the infection levels. In particular, the non-parametric Mann–Whitney test was applied for independent samples. For statistical analysis, the XlStat ver. 2013.2.04—Addinsoft software was used.

## 3. Results and Discussion

The physical response of chickens administered with the *C. jejuni* field strain and/or bacteriophages was indistinguishable from that of the control birds in terms of no apparent loss of appetite, no reduction in weight gain, no alteration in locomotion, and no diarrhea or respiratory distress. Post-mortem examination of major organs (i.e. hearts, livers, and kidneys) did not show any macroscopically appreciable pathological abnormalities in all treated birds. At this stage, the authors did not assess any potential dysbiosis side effects nor the presence of markers for inflammation in the phage-treated animals. Due to the experimental design and the results of the post-mortem analysis, and as supported by the results of previous works [23], it is likely that no adverse effects were potentially present in this study. However, more in-depth analyses will be carried out in future scale-up trials in relation to these specific features. Data on whole-genome sequences from Φ7-izsam and Φ16-izsam could not be reported at the present because this material is still undergoing a patent registration process. However, these two phages satisfied all the preliminary criteria for safety evaluation; i.e. absence of genes coding for AMR, integrases, or toxic factors. The overall results from the in vivo trial are reported in Table 3. In particular, at day 0, no *Campylobacter* was found in the swab samples. At day 10, *C. jejuni* had an average infection load of 10^8^ cfu/mL, demonstrating an effective bacterial colonization in the animals under assessment. At day 30, a partial slaughtering of chickens was necessary for the evaluation of the average *C. jejuni* infection load for the preparation of the experimental phage suspensions to be administered to experimental groups (MOI 0.1 and 1). At this time, the *C. jejuni* colony numbers recovered from the cecal contents were 10^8^ cfu/gr. At day 40, the *C. jejuni* count showed a significant difference between bacteriophage-treated birds (B and C) and control birds (A) throughout the experimental period investigated. In particular, in the lower intestine of birds from control group A, the *C. jejuni* load was 10^8^ cfu/gr, while both treated groups showed the following infection levels: 10^7^ cfu/gr for group B treated with phages at MOI 0.1; and 10^6^ cfu/gr for group C treated with phages at MOI 1. These results showed a total reduction in *Campylobacter* numbers of 1 log_10_ for group B and of 2 log_10_ for group C (Table 3). The non-parametric Mann–Whitney test applied for independent samples revealed that the differences among the *C. jejuni* infection loads from the three experimental groups were statistically significant (*p* < 0.0001).

The bacteriophages assessed in this study were previously isolated by applying a novel protocol based on the use of a broader spectrum of bacteria—a mix of *Campylobacter* strains with somatic and flagella antigenic differences—in order to amplify the chance to detect, in the poultry environment, a more diversified group of lytic phages against *C. jejuni* [29]. Bacteriophages (Φ7-izsam and Φ16-izsam) were chosen for the in vivo trial because of their wide host-range spectrum and for particular virulence activity against AMR *C. jejuni* strains [28]. In particular, these phages showed in vitro lytic activities against *C. jejuni* strains resistant to cyprofloxacin, nalidixic acid, and tetracycline. It is worth noting that the authors previously reported a very interesting and important finding: the loss of resistance to antibiotics in *C. jejuni* strains, verified with the change of the antimicrobial resistance spectrum assessed by antibiotic-susceptibility tests [28]. This feature has been rarely exploited by scientists, and the first evidence was reported by EFSA experts in 2016 [30]; in particular, *Listeria monocytogenes* strains originally resistant to ciprofloxacin and erythromycin were shown to revert to being antibiotic-sensitive after infection with ΦP100.

The authors believe that the timing of phage administration in poultry is also a determinant factor for success in phage applications, because of the high risk of development of phage-resistant bacterial mutants. In this regard, Wagenaar et al. (2005) [26] demonstrated that after the initial 3 log_10_ reduction of *C. jejuni* counts after phage applications, the *Campylobacter* load increased again within five days and plateaued at 1 log_10_ lower than the controls. To avoid this problem and to reduce the time for resistant strains to emerge, Carvalho et al. (2010) [27] proposed a cocktail of phages rather than just one phage, and achieved an approximately 2 log_10_ reduction. Loc Carrillo et al. (2005) [25] demonstrated that increasing the time between treatment and slaughter from 24 h to 2–4 days may reduce the effective decrease of *Campylobacter* spp. counts in the cecal contents of phage-treated broiler chickens. In particular, they demonstrated values of *Campylobacter* reduction levels from 0.5 log_10_ to 5 log_10_ after phage applications in in vivo trials. Kittler et al. (2013) [31] instead reported very variable results, according to the type of phage used (from no evident *C. jejuni* reduction to 3.2 log_10_ cfu/gr lower bacterial counts than in the control). In the present study, to minimize the potential development of *Campylobacter* mutants resistant to phages, the authors administered, for the first time in science, two different bacteriophages, Φ16-izsam and Φ7-izsam, 24 h apart from each other, with the second applied 24 h before slaughtering. Chickens were treated with two different phage MOIs (0.1 and 1), chosen on the bases of the results of previous in vitro efficacy tests [32]. In particular, in our preliminary in vitro results, we already experienced the possible success of this sequential administration of two different phages; in fact, *Campylobacter*’s growth trend not only was maintained at lower levels after the second phage application when compared to the negative (non phage-treated) control, but also in the *Campylobacter* cultures treated with only one phage (Φ16-izsam).

The experimental trial within this research demonstrated that both the MOIs used produced an efficient reduction in *C. jejuni* counts in the chicken guts one day after administration of the second phage suspension. These results suggest the possibility that bacteriophages may be administered shortly before slaughter (phage administrations completed 24 h before slaughtering), subsequently minimizing the risk to disseminate potential phage-resistant *Campylobacter* strains in the farm environment. In the present study, the authors did not assess the development of phage-resistant mutants but this issue will be analyzed more in-depth in future scale up trials. Our results are in line with those reported by the authors previously cited. In particular, we achieved a total reduction in *C. jejuni* numbers of 1 log_10_ for those birds treated with MOI 0.1, and 2 log_10_ for those treated with MOI 1. Nevertheless, opinions diverge regarding which MOI could be the best to apply to obtain the most efficacious results. Tomat et al. (2013) [33] suggested that the application of higher numbers of phages and higher MOI values (>1) will result in a greater reduction of the numbers of the pathogen (passive-mode applications [34]). In the authors’ laboratory experience, the use of low MOIs (≤1) has always been more successful when compared to the use of higher MOIs (>1), both in vitro [32] and in vivo (present study). This evidence supports the “active mode of phage-therapy” as stated by Cairns et al. (2009) [34]: “The phage concentration will not exhibit net growth unless the host concentration is above the appropriate proliferation threshold, and when this occurs phage proliferation becomes relatively rapid until the phage concentration reaches the inundation threshold and begins to suppress the host population”.

The results achieved in this study are very encouraging; in fact, according to the EFSA opinion adopted in 2011 [16], the reduction of *Campylobacter* contamination loads in broiler meat of 1–2 log units would reduce the public health risk for human consumers to contract Campylobacteriosis associated with consumption of chicken meals by up to 90%.

In commerce, no phage-based applications against *Campylobacter* spp. are available at the moment. Nevertheless, since the veterinary use of antibiotics is very controversial because of AMR, phage therapy may become more generally accepted. Our results clearly show that phage therapy is able to control infections in broilers also caused by AMR *C. jejuni* strains. Moreover, as expected [23], the phage administration in chickens showed no detectable adverse effects; e.g. abnormal physical response, loss of appetite, reduction in weight gain, alteration in locomotion, or diarrhea or respiratory distress.

More research and scale-up trials are needed to improve the efficiency of phage therapy, to explore the robustness of this experimental approach when applied to modern chicken-management systems, and to assess the phage efficacy versus different *Campylobacter* field populations in natural breeding conditions. Phages are generally given to animals in feed or water [27] as simple purified viral suspensions or after microencapsulation [35]. In particular, the most effective and suitable method for Φ7-izsam and Φ16-izsam administration in poultry on an industrial scale needs to be explored in more details in the near future. Very recently, some authors have speculated about the cost benefits related to the use of phages in therapy [36]. Though the opinions are very different, in general, high-scale phage productions are of relatively low cost when compared to the production of new classes of antibiotics. Moreover, the efficacy of phage-based formulations has been extensively demonstrated, including few side effects when compared to the large burden related to AMR when over-using some classes of chemicals [13]. This may also illuminate aspects such as the evaluation of the carriage and shedding of *Campylobacter* spp. from infected birds, together with comparisons about the release of *C. jejuni* in the slaughter process, both from phage-treated animals and from naturally infected and non-treated birds.

## 4. Conclusions

*Campylobacter* colonization in poultry poses a serious threat to human health when entering the food chain after slaughtering. The field bacteriophages tested in this research in an in vivo assay were able to assure efficacy by reducing the total load of bacterial contamination by up to 2 log_10_ in experimentally infected birds. In particular, the results were obtained against a specific AMR *Campylobacter jejuni* strain. An innovative two-step phage administration, 24 h before slaughtering, was applied with the use of low phage MOIs (0.1 and 1).

The effective treatment achieved in this research demonstrated that the alternative phage-therapy approach proposed in this in vivo trial to contrast *Campylobacter* in poultry should be deeper investigated in scaled-up field trials, and deserves serious attention for a potential application in the future.

## Figures and Tables

**Table 1 viruses-13-01428-t001:** Optimal temperatures at which the animals were housed at different days of life during the in vivo trial.

Day of Life	Temperature °C
Day 1	33
Day 4	32
Day 7	30
Day 10	29
Day 13	28
Day 16	27
Day 19	25
Day 22	24
Day 25	23
Day 28	22
Day 31	21
Day 34	20
Day 40	20

**Table 2 viruses-13-01428-t002:** Animal feeding conditions during the in vivo trial.

Days of Life	Feed
0–10 days	Starter feed
11–21 days	1st growing feed
22–30 days	2nd growing feed
31–39 days	Finishing feed

**Table 3 viruses-13-01428-t003:** Experimental design of the in vivo trial, including time (T), aim of the action, and results.

Day	Experimental Action	Number of Animals	Aim	Results
T0	Cloacal swab	75	To verify the absence of natural *Campylobacter* spp. colonization	0 cfu/swab
T1	*C. jejuni* administration	75	Experimental infection	
T10	Cloacal swab	75	To verify *C. jejuni* experimental infection	10^8^ cfu/swab
T30	Partial slaughtering	6	Evaluation of *C. jejuni* counts for MOI preparations	10^8^ cfu/gr cecal content
T37	Group construction (A, B and C)	23 per group		
T38	Phage 16 and SM-CaCO_3_ administration	46 treated with phage 16 23 treated with SM-CaCO_3_	Phage therapy	
T39	Phage 7 and SM-CaCO_3_ administration	46 treated with phage 7 23 treated with SM-CaCO_3_	Phage therapy	
T40	Slaughtering	69	Phage therapy	Group A: 10^8^ cfu/gr cecal content Group B: 10^7^ cfu/gr cecal content Group C: 10^6^ cfu/gr cecal content

## Data Availability

All the data presented in this study are available within the article.

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
