# Peer review of "Bacteriophage Therapy to Reduce Colonization of *Campylobacter* *jejuni* in Broiler Chickens before Slaughter"

_viruses, 2021, doi:10.3390/v13081428_

Round 1

Reviewer 1 Report

I found the results interesting and recommend that the paper be published subject to the authors making several minor changes to the text. 

The paper makes a useful contribution to the debate concerning the utility of bacteriophages to reduce the concentration of pathogens in animals before slaughter with the potential of reducing food-related illness. The results are particularly noteworthy in that the animals were infected with a strain of C. jejuni that was resistant to several antibiotics.

The proposal that phages could be administered a short time before slaughter and potentially reduce campylobacter populations to a level that would reduce the no of cases of food poisoning is significant at several levels including cost, animal health, and utility. However, the authors need to be open to further work being required to confirm this assertion.

I note the comments about the phages being the subject of a patent application. While there are references to the authors' previous work a more detailed summary of the properties of the two phages should be given. While ARGs are rarely encoded in phages it would be helpful to indicate that these are absent in the two phages.

The English language usage in the last paragraph of the introduction should be improved to maximize the impact of the paper.

The Introduction would benefit from several sentences giving an overview of intestinal dysbiosis as a potential side effect of phage therapy. I accept that the experimental design used here virtually eliminates this possibility. However, for completeness, the comments should be added.

The comment on the loss of resistance to antibiotics following phage attack is new to me, I was not aware that this is a widely accepted phenomenon? Please add a reference. Did you check your bacterial isolates to see if their antimicrobial resistance spectrum had changed? If not it might be interesting to do this in the future.

Table 3 explaining the experimental design could be improved. If you have the pfu data please provide the phage concentration in the caecal contents. 

 Is placebo the right word when working with animals and using this experimental design? English usage? 

The  MOI results are potentially important  for future research and application of phage therapy.  How precise are the values? I am familiar with the shedding of MAP and salmonella in cows.  In cows, the numbers vary depending on many factors. However,  regardless of the precise value of the MOI in practice the data suggest that higher MOIs are not necessarily the best strategy and this is an important finding.

A brief mention should be made of cost benefits and how the phages would be used in practice. 

A very small point. In section 3 the following sentence is noted: “In particular, in the lower intestine of birds from the control group  A, C. jejuni load was 10^8.6 cfu/gr while both treated groups showed the following infection levels: 10^7 cfu/gr for group B treated with phages at MOI 0.1; 106 cfu/gr for group C, treated with phages at MOI 1.” Why use  10^8.6. This is 398,107,170.553 or 3.98 x 10^8 or approximately 10^8. Why do you not say 10^8 since 10^6  and 10 ^7 are used in the same sentence? 

The authors, rightly, mention the development of phage resistance by host bacteria several times. For completeness, it would be helpful if the authors mentioned (briefly) the strategy they will use when phages cease to work because of the development of host resistance.

Finally, it would be helpful if the authors commented on the likelihood of phage therapy increasing markers for inflammation or changing the diversity of microbial populations in caecal material. Note, I do accept that due to the experimental design it is likely that there will be no adverse effects. However, for completeness, these do need to be mentioned in this paper.

Author Response

I found the results interesting and recommend that the paper be published subject to the authors making several minor changes to the text. 

The paper makes a useful contribution to the debate concerning the utility of bacteriophages to reduce the concentration of pathogens in animals before slaughter with the potential of reducing food-related illness. The results are particularly noteworthy in that the animals were infected with a strain of C. jejuni that was resistant to several antibiotics.

The proposal that phages could be administered a short time before slaughter and potentially reduce campylobacter populations to a level that would reduce the no of cases of food poisoning is significant at several levels including cost, animal health, and utility. However, the authors need to be open to further work being required to confirm this assertion.

We thank the valuable reviewer for his/her comments and for the precious advices given in order to improve the quality of our manuscript.

I note the comments about the phages being the subject of a patent application. While there are references to the authors' previous work a more detailed summary of the properties of the two phages should be given. While ARGs are rarely encoded in phages it would be helpful to indicate that these are absent in the two phages.

Thank you the valuable reviewer for the comment. Some details about phage genome have been introduced in lines 205-207

The English language usage in the last paragraph of the introduction should be improved to maximize the impact of the paper.

Thank you the valuable reviewer for the comment. Improvements have been introduced in lines 100-108

The Introduction would benefit from several sentences giving an overview of intestinal dysbiosis as a potential side effect of phage therapy. I accept that the experimental design used here virtually eliminates this possibility. However, for completeness, the comments should be added.

Thank you the valuable reviewer for the comment. Some details about this topic have been introduced in lines 74-80 and 198-203

The comment on the loss of resistance to antibiotics following phage attack is new to me, I was not aware that this is a widely accepted phenomenon? Please add a reference. Did you check your bacterial isolates to see if their antimicrobial resistance spectrum had changed? If not it might be interesting to do this in the future.

Thank you the valuable reviewer for the comment. This is indeed a very interesting finding that we verified also in our past researches. Reference has been added in line 236-238 and also some details about our previous experience, in line 233-235.

Table 3 explaining the experimental design could be improved. If you have the pfu data please provide the phage concentration in the caecal contents. 

Thank you the valuable reviewer for the comment. Unfortunately we did not assess phage pfu in the caecal contents. This will be a very good advice for next trials.

Is placebo the right word when working with animals and using this experimental design? English usage? 

Thank you the valuable reviewer for the comment. Placebo word was removed all over the text and substituted with (SM-CaCO3) in lines 178 and 184 and in Table 3

The MOI results are potentially important  for future research and application of phage therapy.  How precise are the values? I am familiar with the shedding of MAP and salmonella in cows.  In cows, the numbers vary depending on many factors. However,  regardless of the precise value of the MOI in practice the data suggest that higher MOIs are not necessarily the best strategy and this is an important finding.

Thank you the valuable reviewer for the comment. Our standard deviations in relation to bacterial and phage counts were very little and, since we did not find these values reported in most of the updated papers on phage therapy, we preferred not to insert them and to reduce the values to the most significant ones (i.e. 10^8 – see line 217 and table 3). For this reason, we are confident to say that the MOI values are approx. precise.  

A brief mention should be made of cost benefits and how the phages would be used in practice. 

Thank you the valuable reviewer for the comment. Some details about cost benefits and how the phages would be used in practice have been introduced in lines 298-307.

A very small point. In section 3 the following sentence is noted: “In particular, in the lower intestine of birds from the control group  A, C. jejuni load was 10^8.6 cfu/gr while both treated groups showed the following infection levels: 10^7 cfu/gr for group B treated with phages at MOI 0.1; 106 cfu/gr for group C, treated with phages at MOI 1.” Why use  10^8.6. This is 398,107,170.553 or 3.98 x 10^8 or approximately 10^8. Why do you not say 10^8 since 10^6  and 10 ^7 are used in the same sentence? 

Thank you the valuable reviewer for the comment. As replied for the MOIs, we now reported 10^8 in lines 217 and table 3.

The authors, rightly, mention the development of phage resistance by host bacteria several times. For completeness, it would be helpful if the authors mentioned (briefly) the strategy they will use when phages cease to work because of the development of host resistance.

Thank you the valuable reviewer for the comment. In our study we did not verify the existence of phage-resistance mutants, as we now more clearly stated in lines 268-270. Our actions were addressed to avoid any “potential” phage-resistant mutants development. It will be useful to verify this in our future in-depth studies.

Finally, it would be helpful if the authors commented on the likelihood of phage therapy increasing markers for inflammation or changing the diversity of microbial populations in caecal material. Note, I do accept that due to the experimental design it is likely that there will be no adverse effects. However, for completeness, these do need to be mentioned in this paper.

Thank you the valuable reviewer for the comment. These aspects were introduced in lines 74-80 and 198-203

Reviewer 2 Report

This is a finely written paper on the effect of phages able to control Campylobacter populations in broiled chickens. While being a pity that not DNA sequencing data are presented because of a patent application, this referee feels that the paper should be published in its present form in order to avoid unwanted delays.

Author Response

This is a finely written paper on the effect of phages able to control Campylobacter populations in broiled chickens. While being a pity that not DNA sequencing data are presented because of a patent application, this referee feels that the paper should be published in its present form in order to avoid unwanted delays.

On behalf of all the authors, I wish to refer our gratitude for your valuable comments.

Regards.
